# The cauliflower mosaic virus transmission helper protein P2 modifies directly the probing behavior of the aphid vector *Myzus persicae* to facilitate transmission

**Maxime Verdier**[1]ʘ, **Quentin Chesnais**[1]ʘ*, **Elodie Pirolles**[2], **Stéphane Blanc**[2], **Martin Drucker**[1]*

**1** SVQV UMR 1131 INRAE Centre Grand Est–Colmar, Université Strasbourg, Colmar, France, **2** PHIM, INRAE Centre Occitanie–Montpellier, CIRAD, IRD, Université Montpellier, Institut Agro, Montferrier-sur-Lez, France

ʘ These authors contributed equally to this work.
* quentin.chesnais@inrae.fr (QC); martin.drucker@inrae.fr (MD)

**Data Availability Statement:** All relevant data are within the manuscript and its Supporting Information files.

## Abstract

There is growing evidence that plant viruses manipulate their hosts and vectors in ways that increase transmission. However, to date only few viral components underlying these phenomena have been identified. Here we show that cauliflower mosaic virus (CaMV) protein P2 modifies the feeding behavior of its aphid vector. P2 is necessary for CaMV transmission because it mediates binding of virus particles to the aphid mouthparts. We compared aphid feeding behavior on plants infected with the wild-type CaMV strain Cabb B-JI or with a deletion mutant strain, Cabb B-JIΔP2, which does not produce P2. Only aphids probing Cabb B-JI infected plants doubled the number of test punctures during the first contact with the plant, indicating a role of P2. Membrane feeding assays with purified P2 and virus particles confirmed that these viral products alone are sufficient to cause the changes in aphid probing. The behavior modifications were not observed on plants infected with a CaMV mutant expressing P2Rev5, unable to bind to the mouthparts. These results are in favor of a virus manipulation, where attachment of P2 to a specific region in the aphid stylets–the acrostyle–exercises a direct effect on vector behavior at a crucial moment, the first vector contact with the infected plant, which is essential for virus acquisition.

## Author summary

Some pathogens including plant viruses manipulate vectors to optimize transmission. The manipulations can be indirect meaning that pathogens alter host traits such as color or odor that attract or deter vectors. Other modifications are direct, i.e. uptake of virus compounds changes vector behaviors. Direct effects have been reported for viruses that are internalized by their vectors and interact strongly with the vector from within, for example with the nervous system. Here we show that contact of a virus protein with the vector's exterior mouthparts suffices to induce a direct effect: binding of the non-structural

**Funding:** This work was funded by Agence Nationale de la Recherche grant Rome, grant number ANR-18-CE20-0017-01 (to MD). MV was financed by a Ph.D. fellowship from Université de Strasbourg. The funders had no role in study design, data collection and analysis, decision to publish, or preparation of the manuscript.

**Competing interests:** The authors have declared that no competing interests exist.

cauliflower mosaic virus protein P2 to aphid stylets during test punctures modifies probing activity instantly, thereby facilitating virus transmission. The fact that here no intimate virus-vector interactions are required for vector manipulation and transmission, could explain the broad vector range of CaMV and other non-circulative viruses.

## Introduction

Plant viruses are economically important pathogens and most of them require a vector for transmission [1]. Insects are the most common vectors and, among these, Hemiptera such as aphids, whiteflies, plant- and leafhoppers account for the transmission of the majority of the vector-borne viruses [2,3]. This is likely due to their particular mouthparts, the stylets. The stylets' morphology is adapted to piercing-sucking feeding behavior and allows aphids and other hemipterans to acquire and inoculate viruses into plant tissues with great precision and without inflicting major damage.

A growing corpus of theoretical modelling and empirical research shows that viruses and other parasites modify hosts and vectors to optimize their transmission [4–7]. These modifications may be referred to as parasitic manipulation when two conditions are met. First, the phenotypic changes in the host or vector enhance the fitness of the pathogen and second, they are under the genetic control of the pathogen [8]. Evidence for the first criterion has been cumulated by numerous studies for plant viruses [9,10]. In contrast, the viral factors responsible for changes in vector fitness are often unknown [11]. Plant viruses alter host plant phenotype that in turn influences vector attraction and feeding behavior, and consequently virus acquisition [12–14]. Some viral genes have been implicated in these indirect plant-mediated alterations of vector traits [15–17].

Viruses can–after acquisition by vectors–also alter vector behavior directly. This has been studied in particular for plant viruses relying on the circulative transmission mode. Such viruses traverse the intestine, cycle through the hemocoel and accumulate in the salivary glands, before inoculation as a saliva component into new plant hosts. During their passage, they can interact with various host organs, for example the brain, the salivary glands or antenna and modify vector behavior in ways that are conducive to virus transmission [18–20]. Similar manipulations have also been described for other pathogens that replicate in their vectors [21]. To the best of our knowledge, there is no evidence that non-circulative viruses, i.e. viruses that bind to vector mouthparts for their passage to a new host, can change vector behavior directly, whereas plant-mediated effects are well-documented [22]. One report detected altered feeding behavior on healthy test plants of whiteflies viruliferous with a non-circulative crinivirus, but since the insects were raised for two generations on infected plants before the experiments, plant-mediated effects cannot be excluded [23].

Cauliflower mosaic virus (genus *Caulimovirus*) is transmitted by aphids using the non-circulative mode. CaMV virions are retained in the aphid mouthparts (stylets) by attaching to cuticular proteins (stylins) located at the stylet tip in a zone called the acrostyle [24–26]. Transient adherence of virions occurs via a helper component, the viral protein P2 [27,28]. P2 forms a protein bridge between the stylets and the virions, most likely by binding with its N-terminus to stylins and with its C-terminus to the virion, more precisely to the capsid-associated viral protein P3 (P3:virions) [29–31]. Aphids can acquire the helper component P2 and P3:virions simultaneously or sequentially, i.e. either preformed P2:P3:virions complexes or first P2 and in a second step P3:virions [32].

Aphids landing on a new plant will explore the plant's suitability with test punctures. For this, they insert the stylets in epidermis and mesophyll cells, salivate briefly into the cytoplasm and ingest actively some of the cellular contents. If the plant is susceptible, the stylets advance deeper into the tissue, doing more test punctures until they are inserted into the sieve tubes. Here the feeding behavior changes: after an active salivation phase, the aphids ingest phloem sap passively and continuously, their principal food source. Because P2 locates exclusively in infected cells, it can be acquired only during intracellular test punctures, whereas P3:virions can be acquired during test punctures and during phloem sap ingestion [33].

Our previous work showed that CaMV infection of the model plant *Arabidopsis thaliana* caused *Myzus persicae* aphids to feed longer from the phloem, which might enhance the acquisition of P3:virion complexes from the phloem sap. We demonstrated that the P6-TAV protein of CaMV contributes majorly to this altered feeding behavior [34]. P6-TAV is a multifunctional protein responsible for most CaMV symptoms and modifications of the physiology of the host [35,36]. Therefore, it is most likely that it exercises an indirect host-mediated effect on the behavior of the aphid vector. We were interested to investigate whether a non-circulative virus like CaMV could also encode factors having a direct effect on the vector. We chose P2 to test for this hypothesis, because it contains the interaction domain for binding to the aphid stylets, making it an excellent candidate.

## Results

### Aphid feeding behavior is different on plants infected with wild type CaMV expressing P2

To test for a possible effect of P2 on aphids, we chose to compare the behavior of aphids on turnip plants infected with wild type CaMV isolate Cabb B-JI (JI) or with the CaMV P2 deletion mutant Cabb B-JIΔP2 (JIΔP2). We first verified that the deletion did not affect the infectivity of the virus. No difference in symptoms was observed between plants infected with JI or JIΔP2. All plants displayed characteristic leaf bleaching that initially affected only the veins (mosaics) and that covered later in infection the entire leaf (Fig 1A). Fully infected plants showed, in addition, leaf curling and retarded plant growth. The first symptoms appeared 6 days after mechanical inoculation of plants with the viruses and a day later all plants were symptomatic (S1 Fig). Then, we studied the accumulation of the CaMV proteins P2, capsid protein P4 and P6-TAV by western blot in infected turnips (Fig 1B). As expected, P2 was detected in plants infected with JI, but not in plants infected with JIΔP2. Accumulation of P4 and P6-TAV was similar in JI and JIΔP2-infected turnips. Taken together, the deletion of the P2 coding sequence had no impact on the timing and severity of symptoms, and it did not affect CaMV replication as judged by the accumulation of P4 and P6-TAV. Thus, the experimental setup was suited to compare aphid behavior on infected plants expressing P2 vs. those that did not.

We placed aphids on mock-inoculated, JI and JIΔP2-infected plants and used EPG to evaluate the effect of the infection, and more specifically that of the presence of P2, on acquisition feeding behavior (Fig 1C and D). Aphids spent significantly more time ingesting phloem sap on plants infected with JI than on healthy plants (Fig 1C), consistent with our earlier report [34]. Deleting P2 from the viral genome had no effect on phloem sap ingestion. When analyzing the occurrence of events, the total numbers of stylet penetrations, pathway phases and intracellular test punctures were significantly lower on infected plants than on mock-inoculated ones, with no difference between JI and JIΔP2. However, we observed P2-specific alterations for the number of intracellular punctures during the first probe, which was twice as high on JI-infected plants than on JIΔP2-infected ones or mock-inoculated ones. Another feeding parameter, the number of phloem sap ingestion phases, was significantly lower on JI-

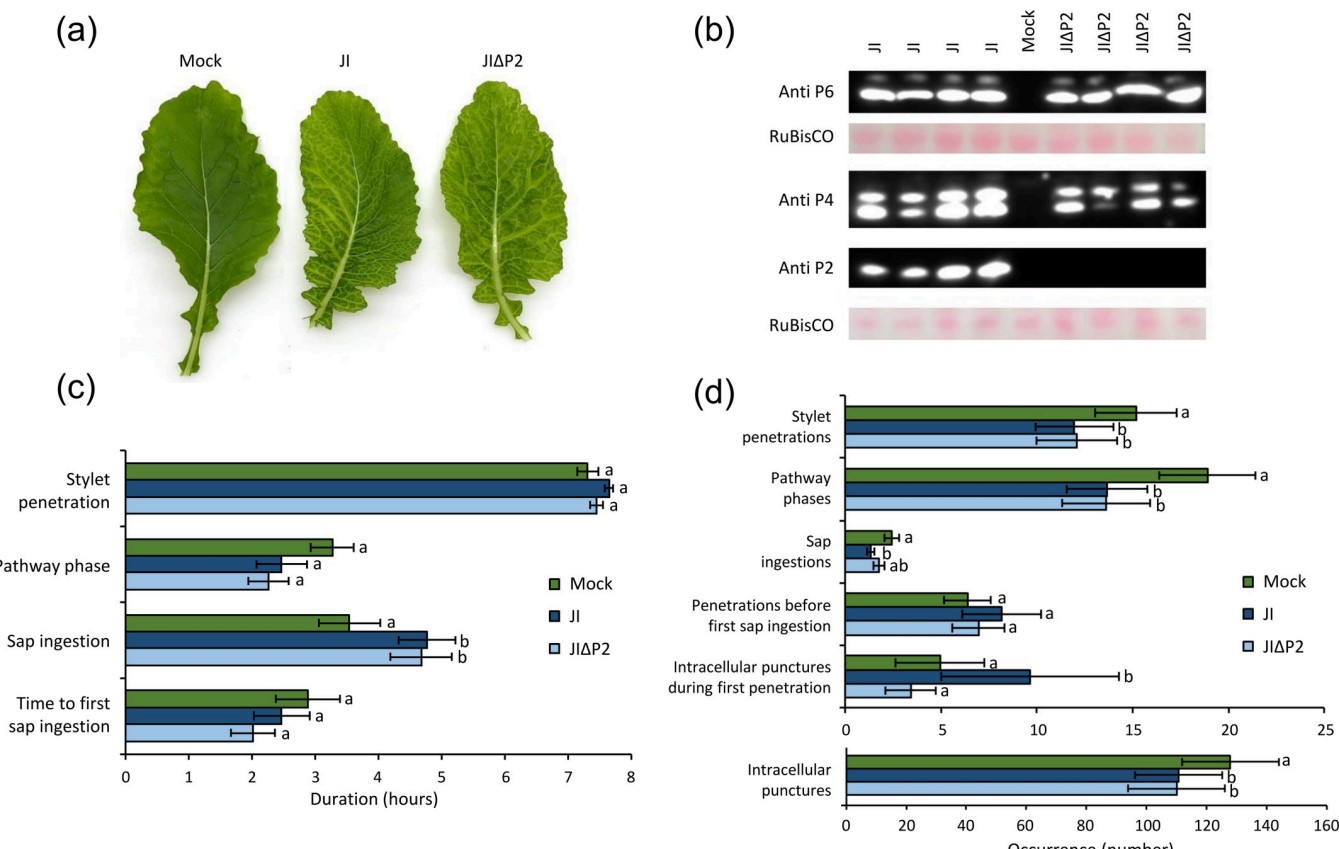

**Fig 1. Feeding behavior of *Myzus persicae* on mock-inoculated, JI- or JIΔP2-infected turnip plants.** (a) Symptoms on turnip leaves at 14 dpi. From left to right, mock-inoculated, JI-infected, JIΔP2-infected leaf. (b) Western blot analysis of the accumulation of P2 (18 kDa), P4 (37, 44 kDa) and P6-TAV (62 kDa) in JI- or JIΔP2-infected turnip plants at 14 dpi. Each lane corresponds to a total protein extract from a different plant. The large RuBisCO subunit is stained by Ponceau S and serves as a loading control. Mock is extract from a mock-inoculated healthy leaf. (c-d) The behavior of individual aphids was recorded by electrical penetration graph (EPG) for 8 h on turnip leaves infected or not with the indicated virus (N = 21–24). Selected EPG parameters are presented sorted according to (c) duration or (d) occurrence. The histogram bars display means and standard errors. Different letters indicate significant differences between plant infection status as tested by GLM (generalized linear model) followed by pairwise comparisons using "emmeans" (p < 0.05 method: Tukey). Statistical analysis of the duration of events indicated significant differences for the duration of phloem sap ingestion on infected vs healthy plants (GLM, Df = 2, $\chi^2$ = 7.776, p = 0.020) but no differences for the total duration of stylet penetration (GLM, Df = 2, $\chi^2$ = 3.868, p = 0.145), the total duration of pathway phase (GLM, Df = 2, $\chi^2$ = 4.037, p = 0.133) and the time to first sap ingestion from phloem (Cox, Df = 2, $\chi^2$ = 0.373 p = 0.185). Statistical analysis of the occurrence of events revealed significant differences for the numbers of stylet penetrations, pathway phases and intracellular test punctures on infected (JI and JIΔP2) vs mock-inoculated plants (GLM, Df = 2, $\chi^2$ = 10.756; $\chi^2$ = 24.948; $\chi^2$ = 37.13, p < 0.001, respectively), and a significant difference for the number of intracellular test punctures during the first stylet penetration on JI-infected plants vs JIΔP2-infected and mock-inoculated plants (0-inflated model, Df = 2, $\chi^2$ = 35.958, p < 0.001).

infected plants than on mock-inoculated turnips, their number on JIΔP2-infected leaves was intermediate (GLM, Df = 2, $\chi^2$ = 6.968, p = 0.031). This indicated a possible, but only partial contribution of P2 to this behavior modification. Taken together, infection with JI and JIΔP2 significantly increased the duration of phloem sap ingestion. Only wild type infection (i.e. presence of P2) doubled the number of intracellular punctures in mesophyll and epidermis during the first stylet insertion, indicating that P2 was associated with this.

## Post-acquisition effect of P2-expressing wild type CaMV on aphid inoculation behavior

The previous experiment indicated modified aphid feeding behavior on infected plants, i.e. during virus acquisition feeding. We wanted to know whether virus infection and P2 also

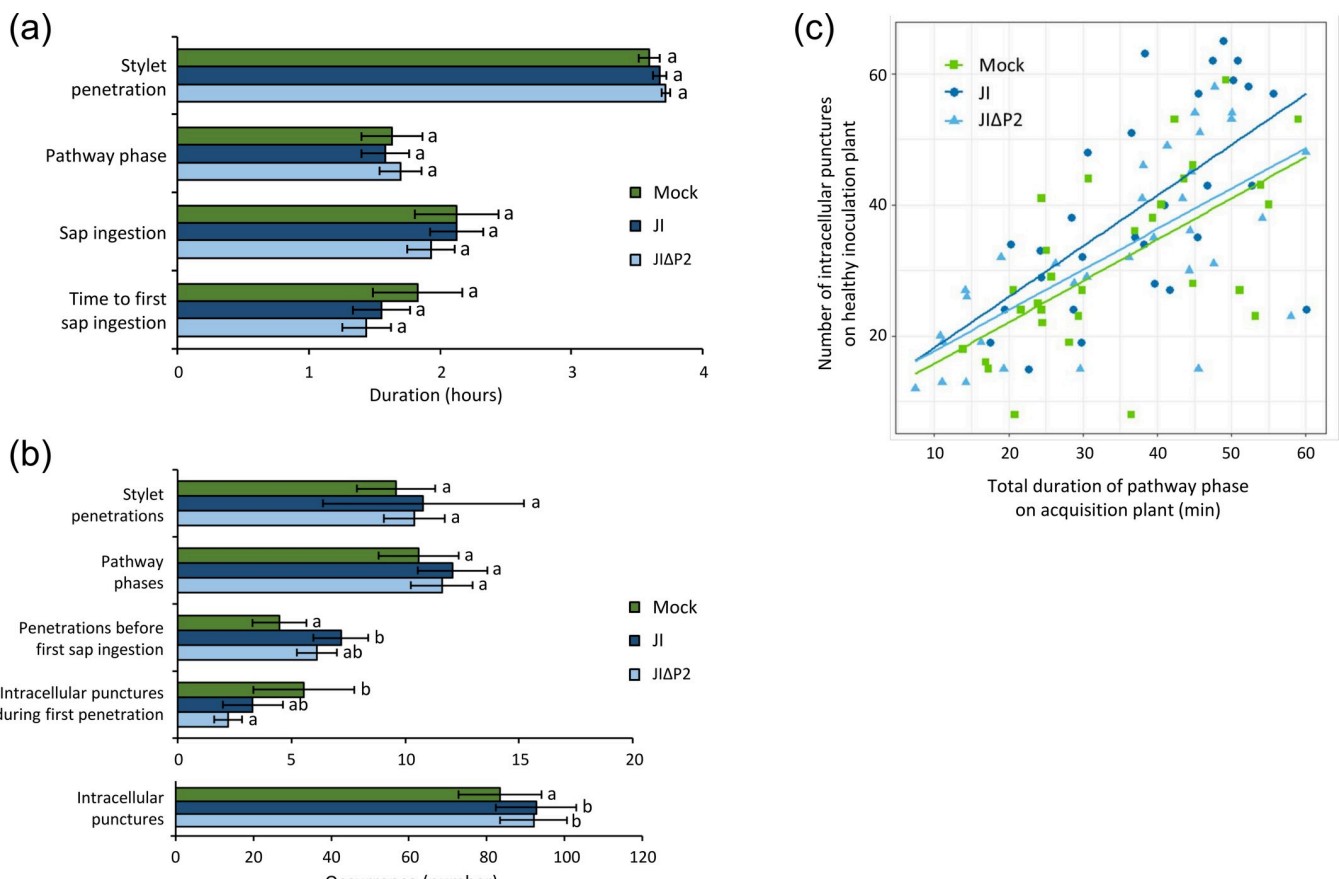

**Fig 2. Feeding behavior during inoculation access period (IAP) of *Myzus persicae* on healthy plants after 1 h acquisition feeding on mock-inoculated, JI-infected or JIΔP2-infected plants (N = 22–26).** (a) presents the duration and (b) the occurrence of behavior phases. The histogram bars show means and standard errors. Differing letters indicate significant differences between plant infection status as tested by GLM (generalized linear model) followed by pairwise comparisons using "emmeans" (p < 0.05 method: Tukey). No significant differences were found for the duration of behavior phases (GLM and Cox models, p > 0.05). For the occurrence of events, significant differences were detected for the number of intracellular test punctures of aphids originating from infected (JI and JIΔP2) vs mock-inoculated plants (GLM, Df = 2, $\chi^2$ = 12.629, p < 0.001), for the number of intracellular punctures during first penetration (0-inflated, Df = 2, $\chi^2$ = 39.740, p < 0.001) and for the number of stylet penetrations before the first phloem sap ingestion for aphids transferred from JI-infected plants vs those transferred from mock-inoculated plants (GLM, Df = 2, $\chi^2$ = 11.353, p < 0.001). (c) Correlation between the number of intracellular test punctures during IAP on healthy plants and the total duration of pathway phase during acquisition access period (AAP) on mock-inoculated (green), JI-infected (dark blue) or JIΔP2-infected (light blue) plants. The coefficients of correlation are r = 0.62, r = 0.61 and r = 0.69 for mock-inoculated, JI-infected and JIΔP2-infected plants, respectively.

modifies feeding behavior post-acquisition, i.e. during the inoculation access period. Aphids were allowed a 1 h acquisition access period (AAP) (under EPG control) on mock-inoculated or infected plants and then transferred to healthy test plants for virus inoculation. The feeding behavior was recorded for another 4 h to assess the impact of viruliferous status on aphid behavior. The total duration of all feeding phases was similar for all conditions (Fig 2A). However, we detected differences in the occurrence of three probing parameters (Fig 2B). The total number of test punctures was significantly higher for aphids transferred from infected plants to healthy plants, compared to those originating from mock-inoculated ones. The number of intracellular punctures during the first stylet penetration was significantly lower for aphids transferred from infected plants compared to those transferred from healthy plants. This was due to infection and not to P2 because there was no difference between JI and JIΔP2 infections. The number of stylet penetrations before the first phloem sap ingestion was elevated for *Myzus persicae* having acquired from JI-infected plants compared to those coming from healthy

plants. Aphids having fed previously on JIΔP2-infected plants required an intermediate number of penetrations until first phloem ingestion. Thus, there may be a tendency for P2 to increase probing events.

To better define a potential role of P2 on the aphid probing behavior, we performed a correlation analysis of the behavior of aphids during the 1 h AAP and the 4 h inoculation access periods (IAP) (Fig 2C). The number of intracellular punctures during IAP on healthy plants correlated strongly with the total duration of the pathway phase during AAP on infected plants (Pearson's correlation; t = 7.988, Df = 90, p < 0.001). Interestingly, the number of intracellular punctures per minute of the pathway phase was similar for aphids coming from healthy plants and JIΔP2-infected plants, while this number was higher for aphids originating from JI-infected plants (LM, $\chi^2$ = 755.500, Df = 2, p = 0.048). This again is in favor of a role of P2 in modifying aphid probing behavior.

## Feeding on purified P2 and virus particles modifies aphid probing behavior

Our results indicated that P2 altered aphid probing behavior on infected plants. This effect could be direct (P2 protein itself changes aphid behavior), indirect via P2-mediated changes in the host plant, or a combination of both. To test for a direct effect of P2, we allowed aphids to acquire recombinant P2 and purified P3:virions (the components of the CaMV transmissible complex) by membrane feeding on artificial medium. Then they were placed on healthy test plants and their feeding behavior was recorded by EPG (Fig 3). This approach eliminated all plant and virus factors that might modify aphid behavior by indirect action of P2. No significant differences for duration of feeding events were observed (GLM, p > 0.05) (S3 Fig). When analyzing the occurrence of events, we found that the number of phloem sap ingestions was not changed by acquisition of P2 (S3 Table). In contrast, the occurrence of several other behavior forms was different. Membrane acquisition of HP2 plus P3:virions increased significantly the total number of stylet penetrations, the number of brief stylet penetrations (< 3 min) (GLM, Df = 4, $\chi^2$ = 19.636, p < 0.001, S3 Table), and the number of stylet penetrations before the first phloem sap ingestion. Further, HP2 plus P3:virions augmented the number of intracellular test punctures and the number of intercellular pathway phases significantly. In general, the effect of HP2 plus P3:virions on aphid behavior was stronger than that of HP2 alone. An exception was the number of test punctures during the first stylet penetration that was significantly enhanced only for HP2.

## Aphids feeding on plants infected with a CaMV mutant that expresses a P2 deficient in stylet binding show mostly normal probing behavior

Our results from the feeding experiment with artificial medium suggest a direct effect of P2 on aphid behavior. At this point, we consider two non-mutually exclusive hypotheses. P2 could modify behavior by binding to the stylets or by interacting with vector factors in the more posterior parts of the digestive tract. To test the first hypothesis, we used the mutant protein P2Rev5, which contains a single Q➜Y mutation at amino acid position 6, which abolishes P2 interaction with the stylets, but maintains all other properties of P2 [37]. Since the original P2Rev5 mutation was characterized in the CaMV CabbS background, we introduced the mutation into the JI genome and obtained the CaMV mutant JI-P2Rev5. Turnip plants infected with JI-P2Rev5 displayed symptoms identical to the wild type-infected plants (Fig 4A). Accumulation of the capsid protein P4 and P6-TAV was identical in JI and JI-P2Rev5-infected plants (Fig 4B). However, P2 accumulation was somewhat lower in JI-P2Rev5-infected plants than in wild type-infected plants (Fig 4B).

Infection with JI-P2Rev5 being similar to wild type virus infection, we assessed aphid behavior on JI-P2Rev5-infected turnip plants. Like in the previous experiment (Fig 1), no

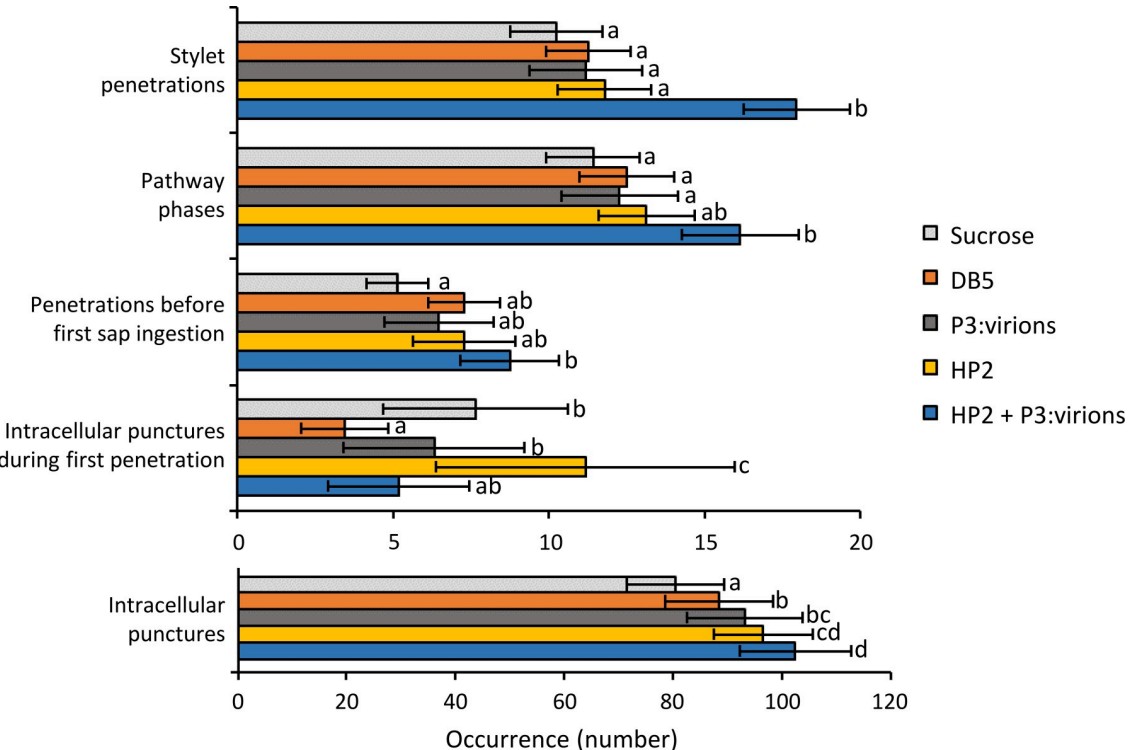

**Fig 3. Feeding behavior of *Myzus persicae* on healthy plants after acquisition of P2 and P3:virions in artificial medium.** The histogram bars display means and standard errors. Before recording aphid feeding behavior on healthy plants, aphids were allowed to feed under electrical penetration graph (EPG) control for 1 h on different artificial media: 15% sucrose in water (light grey); 15% sucrose in DB5 buffer (orange) or 15% sucrose (final) in DB5 buffer supplemented with P3 and purified virus particles (P3:virions, dark grey); his-tagged P2 (HP2, yellow); or HP2 and P3:virions (blue). Purity of the viral components used for aphid feeding assays on artificial medium is shown in S2 Fig. Only aphids having inserted their stylets for at least 5 min in the artificial media were used for the experiments (N = 21–26). Letters indicate significant differences between artificial media as tested by GLM (generalized linear model) followed by pairwise comparisons using "emmeans" (p < 0.05 method: Tukey). Analysis of behavior occurrences indicated a significant effect of HP2+P3:virions on the total number of stylet penetrations (GLM, Df = 4, $\chi^2$ = 23.228, p < 0.001), the number of stylet penetrations before the first phloem phase (GLM, Df = 4, $\chi^2$ = 20.090, p < 0.001), the number of intracellular test punctures (GLM, Df = 4, $\chi^2$ = 72.429; p < 0.001) and the number of pathway phases (GLM, Df = 4, $\chi^2$ = 21.336, p < 0.001). HP2 alone had a significant effect on the number of intracellular test punctures during the first stylet penetration (0-inflated model, Df = 4, $\chi^2$ = 72.430, p < 0.001).

virus effect on the duration of probing events was found (see S4 Fig), so only those related to occurrence are shown here. Aphids on JI-infected plants performed more than twice as many intracellular punctures during the first stylet penetration than on JI-P2Rev5-infected or healthy plants (GLM, 0-inflated model, Df = 2, $\chi^2$ = 77.32, p < 0.001), which is the same result as obtained with JIΔP2 (Fig 1).

The number of aphid stylet penetrations and pathway phases was lower on JI-infected plants than on JI-P2Rev5-infected or healthy plants. This was similar as in Fig 1, but whereas there the differences were statistically significant they were here marginally insignificant (GLM, Df = 2, χ2 = 1.146, χ2 = 5.452, p = 0.059 and 0.065, respectively). The number of penetrations before the first phloem feeding was significantly reduced for JI, but not for JI-P2Rev5, compared to mock-inoculated plants (GLM, Df = 2, χ2 = 13.709, p = 0.001). Except for the total number of intracellular punctures which was slightly (~10%) but significantly elevated for JI-P2Rev5, compared to JI and healthy plants, aphid probing behavior was very similar on JI-P2Rev5-infected and mock-inoculated plants. To sum up, the aphid behavior modifications

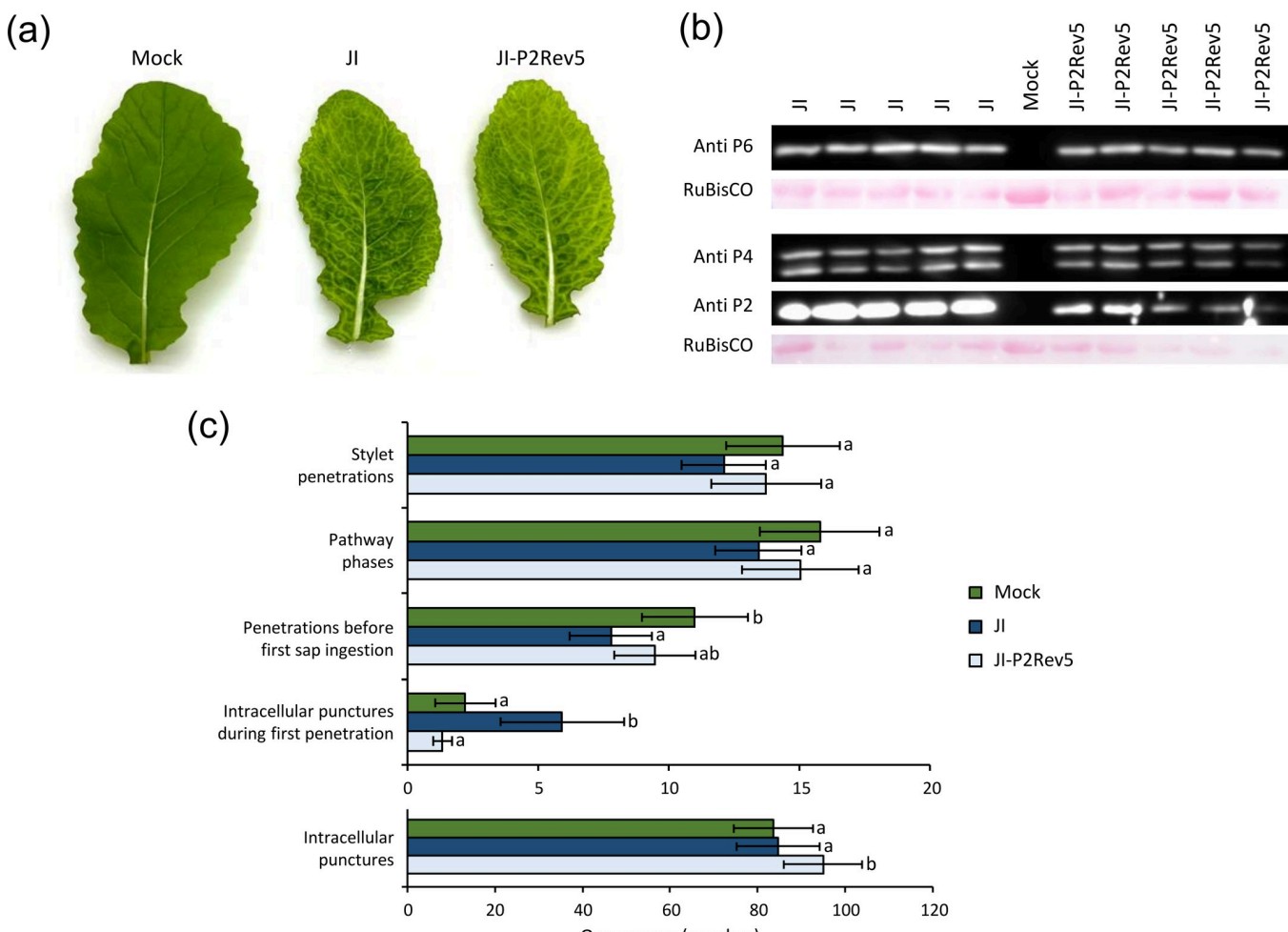

**Fig 4. Probing behavior of aphids on turnips infected with non-transmissible JI-P2Rev5.** (a) Mock-inoculated, JI- and JI-P2Rev5-infected turnip leaves at 14 dpi. (b) Western blot analysis of the accumulation of P2 (18 kDa), P4 (37, 44 kDa) and P6-TAV (62 kDa) in JI- or JI-P2Rev5-infected turnip plants at 14 dpi. Total leaf extracts from 5 infected plants per condition were analyzed. Ponceau S staining of the large RuBisCO subunit is shown as a loading control. Mock, extract from mock-inoculated leaf. (c) Feeding behavior of *Myzus persicae* on mock-inoculated, JI- or JI-P2Rev5-infected turnip plants at 14 dpi. The histogram bars present means and standard errors. The behavior of individual aphids was recorded by electrical penetration graph (EPG) for 4 h (N = 26–28). Different letters indicate significant differences between plant infection status as tested by GLM (generalized linear model), followed by pairwise comparisons using "emmeans" (p < 0.05 method: Tukey). The number of intracellular punctures during the first penetration is significantly higher for JI-infected vs mock- and JI-P2Rev5-infected plants (0-inflated model, Df = 2, $\chi^2$ = 77.32, p < 0.001). The number of penetrations before first sap ingestion is significantly lower for JI-infected compared to mock-inoculated plants (GLM, Df = 2, $\chi^2$ = 13.709, p = 0.001). The total number of intracellular punctures is similar for aphids on JI-infected and mock-inoculated turnips, but elevated for JI-P2Rev5-infected plants (GLM, Df = 2, $\chi^2$ = 23.692, p < 0.001). Statistical analysis revealed no differences for the number of stylet penetrations and pathway phases (GLM, Df = 2, $\chi^2$ = 1.146, $\chi^2$ = 5.452, p = 0.059 and 0.065, respectively).

observed for JI were not found for P2Rev5, in particular the twofold increase in the number of intracellular punctures during the first penetration. The observed behavioral changes indicate that the acrostyle-binding capacity of P2 is important for this. This is supported by the fact that a CaMV isolate harboring the P2Rev5 mutation is not transmissible [37] and that recombinant GFP-tagged P2Rev5 does not bind to dissected stylets in in vitro interaction assays [24]. We set up membrane feeding assays using baculovirus-expressed P2Rev5 to confirm these observations. Unfortunately, the results were not conclusive, probably because we could use for this experiment only crude P2 extracts containing many contaminating proteins and other compounds (S5 Fig; S5 Table).

## Discussion

### CaMV infection alters the overall aphid feeding behavior, while the P2 helper component modulates exclusively aphid probing behavior

Analysis of feeding behavior on infected plants showed significant modification of three feeding activities that are directly related with CaMV acquisition (Fig 1). One, the duration of phloem ingestion increased, two, the total number of test punctures during the entire eight hour observation period decreased, while, three, their number during the first stylet penetration doubled.

Prolonged phloem ingestion might be advantageous for CaMV transmission because the phloem sap contains virus particles and uptake of CaMV from the phloem has been reported before [33]. Prolonged phloem ingestion was observed previously with the same CaMV isolate on two other hosts, *Arabidopsis thaliana* and *Camelina sativa* [34]. This could therefore highlight potential adaptive virus effects. The second observation, reduction of intracellular test punctures over the entire observation period, might be considered counterproductive to acquisition because it should decrease chances to acquire P2, which is absent from the sieve tubes. However, the work by Palacios and coworkers showed that combined uptake of CaMV from tissue and phloem sap might even be more efficient than uptake from epidermis and mesophyll cells alone. Thus, potential negative effects of decreased intracellular test punctures seem to be outweighed by earlier and longer phloem ingestion. This differentiates CaMV from typical non-circulative viruses that are acquired during test punctures in the tissue and lost rapidly when vectors reach the phloem (see for example [38]). These two parameters, phloem ingestion and total number of test punctures, were altered similarly for aphids feeding on plants infected with JI and with JIΔP2. This demonstrates that P2 plays no role in this and that other viral factors are involved. A candidate is P6-TAV, which contributes to the modification of aphid feeding behavior although it alone is not sufficient to induce all behavior changes [34].

The third parameter, the number of test punctures during the first stylet penetration, was doubled on plants infected by JI, but not on healthy plants or those infected by JIΔP2. As removing the coding sequence of P2 from the CaMV genome did not affect other parameters of the infection (Fig 1), we suggest that the increase in test punctures during the first penetration may be caused exclusively by P2. Test punctures are mandatory for acquisition of P2 (which is found only in epidermis and mesophyll cells [33]) and their increase during the first stylet penetration might speed up P2 acquisition and consequently the acquisition of P3:virions. Virus acquisition during the first stylet penetration is also favorable for virus transmission by non-host aphids that reject incompatible plants after the first test punctures and leave them [39,40]. Carmo-Sousa et al. [41] reported similar results for the non-circulative cucumber mosaic virus (CMV). They observed that aphids exhibited during the first 15 minutes–but not later–twice as many test punctures on CMV-infected plants than on mock-inoculated plants. Therefore, this change in aphid behavior during its first contact with the infected plant may be very important for the acquisition of viruses transmitted in a non-circulative manner like CaMV and CMV. An interesting point is that the behavior change is in both cases immediate since it is observed from the very first test punctures onwards. This suggests that the 'active compound' does not need to traverse the digestive tract to induce behavior modification.

We also studied aphid inoculation behavior. We found no differences in phloem feeding behavior of *M. persicae* on healthy test plants, no matter whether the aphids had fed before on mock-inoculated, JI-infected or JIΔP2-infected plants. This is expected because the phloem feeding behavior is essentially influenced by plant quality and phloem sap composition [42], and this is similar for all healthy inoculation plants. However, some relevant probing parameters were modified. First, the total number of test punctures was higher for aphids coming

from infected plants compared to those from mock-inoculated ones. There was no difference between JI and JIΔP2, indicating that not P2 but other viral or plant factors increased the number of test punctures. Second, the number of stylet penetrations before the first phloem sap ingestion was higher when aphids had previously fed on JI-infected plants compared to mock-inoculated plants, whereas this number was intermediate after acquisition feeding JIΔP2-infected plants. P2 might therefore be partially involved in the modification of this parameter. Furthermore, the correlation between the number of intracellular punctures on healthy inoculation plants and the duration of the pathway phase on infected JI but not JIΔP2 acquisition plants (Fig 2C) suggests that P2 (i.e. JI-infected turnips) alters subsequent aphid probing behavior. The increased number of probes and intracellular punctures on a healthy plant by CaMV-carrying aphids might increase the chances for successful inoculation and be beneficial for transmission.

## Evidence for a direct effect of P2 on aphid behavior

P2 might directly or indirectly change aphid probing behavior. Our membrane feeding assays present evidence for a direct effect of P2. Aphids having acquired different combinations of purified P2 or P3:virions through feeding on the artificial medium (thus excluding any interference from other plant or virus factors), showed altered subsequent probing behavior, compared to aphids that had access to artificial medium without viral compounds (Fig 3). Compared to inoculation feeding on healthy test plants after CaMV acquisition from infected plants (Fig 2), inoculation feeding after membrane feeding changed more behavioral parameters. We propose that this is due to the unphysiological conditions of the artificial feeding medium (high salt content, presence of detergent), necessary to maintain P2 active [43]. Therefore, although the membrane feeding experiments do show a direct effect of P2 on aphids, interpretation of the altered behaviors is difficult.

## Hypotheses on the mode of action of P2

How could P2 change aphid probing behavior? P2 binds to a specific region in the stylets, the acrostyle in the common canal in the stylet tips [24,25,44]. The acrostyle at the tip of the maxillary stylets is very restricted in size (~0.2 μm x 5 μm) and located in a stylet region whose diameter does not exceed ~0.5 μm [25]. Binding of P2 and especially of P3:virions (about 35–60 nm in diameter [45,46]) might cause steric hindrance and impede the flows in the common channel during the active ingestion phases (i.e. intracellular test punctures [47]). The changed probing behavior might be an effort of the aphid to compensate for this. In favor of this hypothesis is that membrane acquisition of P2 plus P3:virions had globally a stronger effect on aphid probing behavior than P2 alone. Furthermore, P2 is known to form long paracrystalline filaments that even alone could induce some alteration of the flux in the common canal [43,48]. We observed no effect of P2 on phloem ingestion. An explanation is that this feeding behavior is passive and driven by the high hydrostatic pressure of the phloem sap, which might outweigh steric hindrance effects [49]. An alternative, but not mutually exclusive hypothesis is that binding of P2 or transmissible complexes to the acrostyle affects stylet proteins. The acrostyle is covered with cuticular proteins entangled with the chitin fibers of the stylets [50]. Two of these proteins (called stylins) have been identified and at least one of them–Stylin-1 –can interact with P2 [26]. The binding of P2 to stylins might interfere with their natural function, which is yet unknown. The lack of nerve cells in the maxillary stylets harboring the acrostyle makes it implausible that stylins are signaling receptors and that P2 binding is perceived as a signal [51]. Instead, P2 might displace or compete with attachment of natural stylin ligands–for example saliva effectors [52]–thus making foraging for the aphid more difficult and resulting in an

increased number of test punctures [53,54]. It is interesting to note that the binding of P2 to the acrostyle and the concomitant changes in aphid behavior is evidence that the acrostyle is needed for proper probing activity. P2 might also exercise an effect by binding further down in the digestive tract. However, we believe this is unlikely because aphids on plants infected with the JI-P2Rev5 mutant, which abolishes interaction with the acrostyle, displayed similar behavior as those on mock-inoculated or JIΔP2-infected plants. In particular, they did not show the twofold increase in intracellular punctures during the first stylet penetration (compare Figs 1D and 4C). Taken together, aphid behavior on JIΔP2- and JI-P2Rev5-infected plants seems to show that the interaction of P2 with the acrostyle stylin increases frequency of some characteristic aphid prob-ing behaviors facilitating an efficient virus acquisition.

## Concluding remarks

To our knowledge, a direct and immediate effect on vector behavior of a protein from a virus or other pathogen by simple binding to the latter's mouthparts has not been reported before. The protozoan parasite *Leishmania* infects the gut of its sand fly vector and secretes a gelling protein that together with promastigotes obstructs and damages the mouthparts and gut, incit-ing the sand fly to increase biting frequency on mammal hosts and with it transmission [55]. A similar 'plugging' phenomenon is observed for transmission of the bacterium *Yersinia pestis* by flea vectors [56]. However, there are significant differences to CaMV. First, *Leishmania* and *Yersinia* infect their vectors and establish rather intimate relationships with them, contrary to non-circulative CaMV. Second, the *Leishmania* and *Yersinia* gelling proteins are produced on-site in the intestine, whereas P2 production takes place exclusively in the plant host and seems to affect vector behavior only during and shortly after the vector-host encounter. Thus, the 'site of assault' on the vector is distinctly different in the CaMV pathosystem. It would be inter-esting to explore whether other plant and animal viruses and other pathogens modify vectors directly through host-expressed pathogen factors, as this could be a means to broaden vector specificity since the intimate interactions to prepare subsequent transmission (in this case allo-cation of P2) take place in the host and not in the vector (as for circulative viruses). This type of manipulation would appear particularly relevant for non-circulative viruses such as CaMV that are transmitted generally by numerous aphid vector species (at least 27 for CaMV, [57]). This could impact transmission biology and related fields.

## Materials and methods

### Plant growth and virus inoculation

Turnip seeds (*Brassica rapa* L. var. "Just Right") were provided by Takii Europe B.V. (de Kwa-kel, Netherlands), sown in TS 3 fine substrate (Klasmann-Deilmann, Geeste, Germany) in pots (70 mm x 70 mm x 65 mm) and cultivated at 8 h light / 16 h dark photoperiod at 20 ± 1˚C. Plants were inoculated at the first true leaf stage (9-day-old) and then grown under 14/10 h light/dark cycle at 20 ± 1˚C. Plants were used for experimentation 14 ± 2 days post-inoculation (dpi), when they showed clear symptoms. Initial mechanical inoculation was performed with infectious plasmids. Subsequent inoculations were mechanical and used plant extracts pre-pared from infected turnips. For this, 1 g of infected leaves (21 dpi) were ground with 1 ml 10 mM HEPES pH 7.2 and carborundum and rub-inoculated on 9-day-old turnip seedlings.

### Infectious plasmids

Infectious plasmids for initial inoculation of turnip plantlets were pGreen-35S-B-JI [58] and pGreen-35S-B-JI-ΔP2 [59] that encode the CaMV Cabb B-JI wild type sequence [60], called JI

in the text, and a mutant virus sequence where the P2 sequence is deleted and replaced by an *Apa*I restriction site (referred to as JIΔP2 in the text), respectively, both under control of the 35S promoter. For construction of pGreen-35S-B-JI-P2Rev5, containing a Q➜R mutation of amino acid 6 of P2 [37] the P2 sequence of JI was PCR-amplified with Q5 polymerase (NEB, Evry, France) with primers 5'-AGA<u>GGGCCC</u>ATGAGCATTACGGGT**TAC**CCGCATG-3' and 5'-TTA<u>GGGCCC</u>CTTAGCCAATAATATTCTTTAATCC-3' containing the P2Rev5 mutation (in bold) and 5' and 3' *Apa*I restriction sites (underlined). The amplicon was gel-purified (Machery-Nagel, Hoerdt, France) digested with *Apa*I (NEB) and ligated with T2 ligase (Promega, Charbonnières-les-Bains, France) into pGreen-35S-B-JI-ΔP2 cut with the same restriction enzyme and gel-purified. *Escherichia coli* XL10-Gold were transformed with the ligation product, recombinant colonies identified by colony PCR, and the P2 sequence verified by Sanger sequencing.

## Aphid rearing

The *Myzus persicae* (Sulzer) (Hemiptera: *Aphididae*) clone was originally isolated in the Netherlands. Aphids were reared on Chinese cabbage (*Brassica rapa* L. *pekinensis* var. "Granaat") in a growth chamber at 20 ± 1°C and a 14/10 h light/dark photoperiod.

## Aphid feeding behavior

The electrical penetration graph DC-system (EPG) was used as described by [61] to investigate the effects of CaMV infection on the feeding behavior of *M. persicae*. To integrate one aphid and one plant into an electrical circuit, a thin gold wire electrode (12.5 μm diameter and 2 cm long) was attached with water-based silver glue to the dorsum of an adult apterous aphid that had been immobilized on a 10 μl pipette tip by applying a slight negative air pressure with a vacuum pump. Eight aphids were connected to the Giga-8 DC-EPG amplifier (EPG Systems, Wageningen, Netherlands) and each one was placed directly on the adaxial leaf surface of an individual turnip plant. A second copper rod electrode was inserted into the soil of each potted plant to close the electrical circuit. For the EPG experiments "Acquisition feeding experiment" and "JI-P2Rev5 experiment", the recordings were performed continuously for 8 h and 4 h respectively, during the photophase inside a Faraday cage at 21 ± 1°C. In the second EPG experiment ("Inoculation feeding experiment"), aphids' probing and feeding behaviors were recorded two times. First, aphids were allowed a 1 h acquisition access period on a test plant (either mock-inoculated, JI-infected or JIΔP2-infected). Then the aphid (still attached to the gold wire) was moved to a healthy plant for a 4 h inoculation access period. In the third EPG experiment ("Artificial medium experiment"), aphids were first allowed to feed on an artificial medium during a 1 h acquisition access period and then moved onto a healthy plant for a 4 h inoculation access period. For this setup, the second electrode (copper wire) was inserted in 20 μl medium contained in a sachet formed by two Parafilm membranes spanned over a plastic ring. The feeding medium consisted of 15% sucrose in water or 15% sucrose in DB5 buffer, to which virus components were added as indicated. Acquisition and analysis of the EPG waveforms were carried out with PROBE 3.5 software (EPG Systems). Relevant aphid behavior EPG parameters were calculated with EPG-Calc 6.1 software [62] and were based on the different EPG waveforms described by Tjallingii and Hogen Esch [47]. Aphids that produced signals (i.e. total duration of stylet penetration) for less than 5 h out of 8 h in the first EPG experiment (or 2.5 h out of the 4 h recordings in the second, third and fourth EPG experiments) were excluded from the analysis. This criterion was set at 30 min for the AAP duration on plants whereas for aphids on artificial medium, the threshold was 5 min. For an example of an EPG waveform for aphids on a plant or artificial medium, see S6 and S7 Figs.

## Statistical analysis

The proportion of plants expressing symptoms was analyzed using a Pearson's chi-squared test with Yates's correction ($p < 0.05$). Data on the number of days until the appearance of symptoms on infected turnip plants was not normally distributed. Therefore, we used a non-parametric Wilcoxon rank-sum test ($p < 0.05$).

We used Generalized Linear Model (GLM) with the likelihood ratio and the chi-squared ($\chi^2$) test to determine a statistically significant difference for EPG data. As feeding duration parameters were not normally distributed we used GLM using a gamma (link = "inverse") distribution. Because of the large number of 0's for the "Intracellular punctures during the first penetration" parameter, we used the "zeroinfl" function based on a zero-inflated Poisson model (R package: "pscl"). Parameter "time to first sap ingestion" was modelled using the Cox proportional hazards (CPH) model and we treated cases where the given event did not occur as censored. The assumption of the validity of proportional hazards was checked using the functions "coxph" and "cox.zph", respectively (R packages: "survival" and "RVAideMemoire"). When a significant effect was detected, a pairwise comparison using estimated marginal means (R package "emmeans"; p-value adjustment with Tukey method) at the 0.05 significance level was used to test for differences between treatments (p-values are shown in S6 Table). A total of 28 EPG parameters were calculated (S1–S5 Tables).

Correlation between EPG parameters from inoculated or healthy plants was carried out with a Linear Model (LM). A pairwise comparison using estimated marginal means (R package "emmeans"; p-value adjustment with Tukey method) at the 0.05 significance level was used to test for differences between the three treatments. The coefficient of correlation ("r") was calculated by the function cor.test.

The application conditions of all LM and GLM were verified by inspecting residuals and QQ plots. All statistical analyses were performed using R software v. 4.0.5 (www.r-project.org/).

## Western Blot analysis of infected plants and artificial feeding media

Total protein extracts were prepared from leaves as described previously [34], separated by 15% (P2 and P4) or 12% (P6-TAV) SDS polyacrylamide gel electrophoresis under reducing conditions and transferred onto nitrocellulose membranes as described in [63]. Western blots were performed using antisera raised against P2, P4 and P6-TAV (all diluted 1:2,000, [34]). Secondary antibodies were horseradish peroxidase conjugates, which were used at a 1:10,000 dilution. The same blot was cut into two stripes to test simultaneously for P2 and P4. Bound antibodies were revealed by enhanced chemiluminescence using a G-Box.

## Purification of recombinant proteins and virions

N-terminal his-tagged P2 (HP2) was expressed in baculovirus-infected *Sf9* cells as described previously [43]. Cells from three 75 cm$^2$ cell culture flasks were harvested by centrifugation for 5 min at 500 g and lysed in 9 ml DB5 buffer (50 mM HEPES pH 8.0, 500 mM Li$_2$SO$_4$, 0.5 mM EGTA, 0.2% CHAPS) supplemented with SigmaFast Protease Inhibitor (EDTA-free) and frozen at -80°C until purification. For purification, the thawed cell lysate was centrifuged for 20 min at 24,000 g and the supernatant charged on a column loaded with 300 μl Ni-NTA resin (Macherey-Nagel) pre-equilibrated with DB5. The column was washed with 5 ml DB5 supplemented with 25 mM imidazole. HP2 was eluted with DB5 supplemented with 250 mM imidazole, the protein-containing fractions combined and the imidazole removed by gel filtration with a Sephadex G25 column. Purity and concentration of HP2 were estimated by Instant Blue staining of gels after SDS-PAGE, using BSA as standard.

Wild type P3 was purified from *E. coli* BL21 cells as described [46]. Briefly, cells induced with 1 mM IPTG for 4 h were harvested by centrifugation for 15 min at 4,000 g, washed once with PBS and the pellets were frozen and stored at -80˚C. Cells were lysed by ultrasonication in PBS pH 8 supplemented with 15% glycerol, 0.2 mM DTT, 0.1% Tween20 and SigmaFast Protease Inhibitor (EDTA-free) and centrifuged for 10 min at 18,000 g. The supernatant was heated for 10 min at 65˚C and insoluble proteins were removed by centrifugation for 15 min at 18,000 g. Finally, P3 was purified by differential ammonium sulfate precipitation from 25–40% saturation and ammonium sulphate removed by gel filtration with Sephadex G25 or ultra-filtration with a Vivaspin column.

CaMV particles were purified essentially following the protocol of Gömec described previously [64]. One hundred grams of infected turnip leaves were homogenized in two volumes of phosphate buffer (0.5 M $KH_2PO_4$ pH 7.2, 7.5 g/l $Na_2O_3$) and filtered through four layers of cheesecloth and one layer of Miracloth. Urea and Triton X-100 were added to final concentrations of 1 M and 2.5%, respectively, and the sap was stirred overnight at 4˚C. Then the liquid was clarified by centrifugation for 10 min at 5,000 g and the supernatant was centrifuged for 70 min at 110,000 g. The pellets were resuspended overnight at 4˚C in 12 ml 10 mM HEPES pH 7.2. After centrifugation for 5 min at 10,000 g, the supernatants were loaded on 10–40% sucrose gradients in water and centrifuged for 3 h at 100,000 g in a swing-out rotor. The whitish band visible in the gradients by transillumination was collected with a Pasteur pipette, diluted 1:3 with water and centrifuged for 70 min at 110,000 g. The pellets containing the virus were resuspended in 10 mM HEPES pH 7.2.

## Supporting information

**S1 Fig. Kinetics of symptom onset on turnip plants.** Kinetics of symptom onset on turnip plants after mechanical inoculation with leaf extracts revealed no significant differences between JI and JIΔP2, neither for the percentage of infected plants (N = 29–30) (Pearson's Chi-squared test, $\chi^2$ = 0.01, Df = 1, p = 0.92) nor the day of symptom onset (Wilcoxon rank sum test, W = 437, p = 0.970).
(PDF)

**S2 Fig. Viral components used for aphid feeding assays on artificial medium.** (a) Instant Blue stained protein gel after SDS-PAGE. The slots were loaded with the indicated components. (b) Western blot analysis of purified recombinant his-tagged P2 (HP2), partially purified recombinant P3, and purified virus particles. The blots were developed with the indicated antisera.
(PDF)

**S3 Fig. Feeding behavior of *Myzus persicae* on healthy plants after membrane acquisition of P2 and P3:virions.** Bars show means and standard errors. Before recording aphid feeding behavior on healthy plants, aphids were allowed to feed under EPG control for 1 h on different artificial media: 15% sucrose in DB5 buffer (light grey); DB5 buffer alone (orange) or DB5 buffer supplemented with P3 and purified virus particles (P3:virions, dark grey); his-tagged P2 (HP2, yellow); or HP2 and P3:virions (blue). Viral components used for aphid feeding assays on artificial medium are shown in S3 Fig. Only aphids having inserted their stylets for at least 5 min in the artificial media were used for the experiments (N = 21–26). EPG parameters related to duration are displayed. Letters indicate significant differences between artificial media as tested by GLM (generalized linear model) followed by pairwise comparisons using "emmeans" (p < 0.05 method: Tukey).
(PDF)

**S4 Fig. Feeding behavior of *Myzus persicae* on mock-inoculated, JI- or JI-P2Rev5-infected turnip plants at 14 dpi.** Bars show means and standard errors. The behavior of individual

aphids was recorded by electrical penetration graph (EPG) for 4 h (N = 26–28). EPG parameters related to duration are displayed. Different letters indicate significant differences between plant infection status as tested by GLM (generalized linear model) followed by pairwise comparisons using "emmeans" (p < 0.05 method: Tukey).
(PDF)

**S5 Fig. Effect of recombinant P2Rev5 on aphid behavior.** Bars show means and standard errors. (a-b) Feeding behavior of *Myzus persicae* on healthy plants after membrane acquisition of wild type P2 and P3:virions (P2+P3:virions), P2 carrying the mutation Rev5 and P3:virions (P2Rev5+P3:virions) or of an irrelevant protein (CLINK). Before recording aphid feeding behavior on healthy plants, aphids were allowed to feed under EPG control for 1 h on the different artificial media. Only aphids having inserted their stylets for at least 5 min in the artificial media were used for the experiments (N = 29–34). Selected EPG parameters are presented sorted according to (a) duration or (b) occurrence. Different letters show significant differences between treatments as tested by GLM (generalized linear model) followed by pairwise comparisons using "emmeans" (p < 0.05 method: Tukey). (a) Statistical analysis of the duration of events revealed no difference for parameters shown (GLM, p > 0.05). (b) Statistical analysis of the occurrence of events revealed significant differences for the numbers of intracellular punctures during first penetration and total intracellular punctures (GLM, Df = 2, $\chi2$ = 95.247, $\chi2$ = 49.683, p < 0.001, respectively). None of the other statistically processed parameters showed a significant difference between treatment (see S5 Tables). (c) Instant Blue stained protein gel after SDS-PAGE of Sf9 crude extracts used for aphid feeding assays on artificial medium. Black arrows point to the position of P2 and P2Rev5, P2:GFP and P2Rev5:GFP, and CLINK, respectively. P2:GFP and P2Rev5:GFP were not used in our experiment.
(PDF)

**S6 Fig. Example of a typical EPG waveform recorded on a leaf.** The aphid inserted the stylets into tissue after a few seconds (red arrow) and did probing during the rest of the record (red line). In chronological order, behaviors recorded were a pathway phase (dark grey line) with interspersed intracellular test punctures (greens arrows), then salivation into the phloem (medium grey line), and finally a long period of passive phloem sap ingestion (light grey line).
(PDF)

**S7 Fig. Typical EPG waveform recorded during membrane feeding.** At each insertion of the aphid stylets into the artificial medium a signal was observed (red line). Thus, the duration of presence or absence of aphid stylets in the medium could be measured but no behavior phases could be discerned. This might have been due to the high conductivity of the DB5 buffer that was used as a medium.
(PDF)

**S1 Table. List of 28 EPG parameters statistically processed for the dataset "Acquisition feeding experiment".**
(PDF)

**S2 Table. List of 28 EPG parameters statistically processed for the dataset "inoculation feeding experiment".**
(PDF)

**S3 Table. List of 28 EPG parameters statistically processed for the dataset "artificial medium experiment".**
(PDF)

**S4 Table. List of 28 EPG parameters statistically processed for the dataset "JI-P2Rev5 experiment".**
(PDF)

**S5 Table. List of 28 EPG parameters statistically processed for the dataset "recombinant P2Rev5 experiment".**
(PDF)

**S6 Table. List of p-values of different pairwise comparisons performed when a significant effect was detected with GLM (p-value adjustment with Tukey method at the 0.05 significance level).**
(PDF)

**S1 Text. Production of recombinant P2Rev5 and CLINK.**
(PDF)

## Acknowledgments

We thank Claire Villeroy for aphid rearing and Takii Europe seed company for gracefully providing turnip seeds. We are also thankful to Maria Dimitrova (IBMP Strasbourg) for initial plant inoculation and to Marilyne Uzest (PHIM Montpellier) for critical reading of the manuscript. Part of the plants were provided by the experimental unit of INRAE Grand Est-Colmar (UEAV).

## Author Contributions

**Conceptualization:** Maxime Verdier, Quentin Chesnais, Martin Drucker.

**Formal analysis:** Maxime Verdier, Quentin Chesnais.

**Funding acquisition:** Martin Drucker.

**Investigation:** Maxime Verdier, Quentin Chesnais.

**Methodology:** Maxime Verdier, Quentin Chesnais, Martin Drucker.

**Project administration:** Martin Drucker.

**Resources:** Elodie Pirolles, Stéphane Blanc.

**Supervision:** Quentin Chesnais, Martin Drucker.

**Validation:** Maxime Verdier, Quentin Chesnais.

**Visualization:** Maxime Verdier.

**Writing – original draft:** Maxime Verdier, Quentin Chesnais, Martin Drucker.

**Writing – review & editing:** Maxime Verdier, Quentin Chesnais, Stéphane Blanc, Martin Drucker.

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
