## [Decision Letter · Decision Letter 0]

4 Jan 2023

Dear Dr Drucker,

Thank you very much for submitting your manuscript "The transmission helper protein of a plant virus modifies directly the probing behavior of an aphid vector to facilitate transmission" for consideration at PLOS Pathogens. As with all papers reviewed by the journal, your manuscript was reviewed by members of the editorial board and by several independent reviewers. The reviewers appreciated the attention to an important topic. Based on the reviews, we are likely to accept this manuscript for publication, providing that you modify the manuscript according to the review recommendations.

This is a remarkably interesting study that will make an important and valuable contribution to our understanding of viral "manipulation" of plant host-aphid vector interactions, and to plant virus epidemiology. The three Referees have made a significant number of comments and suggestions for improvement. However, these can be addressed and the revision of the manuscript can be achieved through textual changes and without additional experimentation.

Sincerely,

John P Carr

Guest Editor

PLOS Pathogens

Shou-Wei Ding

Section Editor

PLOS Pathogens

Kasturi Haldar

Editor-in-Chief

PLOS Pathogens

orcid.org/0000-0001-5065-158X

Michael Malim

Editor-in-Chief

PLOS Pathogens

orcid.org/0000-0002-7699-2064

Reviewer Comments (if any, and for reference):

Reviewer's Responses to Questions

**Part I - Summary**

Reviewer #1: This is a well researched and written manuscript. The authors provide evidence that the helper component (P2 protein) of a plant virus (CaMV) can directly modify the probing behaviour of its aphid vector. The authors make use of electrical penetration graph (EPG), wild type virus and a mutant form that doesn’t produce the P2 protein to show that aphid probing and feeding actions associated with virus acquisition and successful inoculation are enhanced in the presence of P2 protein. These findings are significant in that they demonstrate that initial contact between an aphid vector and the virus results in behavioural changes in the aphid and that these changes can be caused by the direct binding of P2 to receptors in the aphid stylet. Further, the authors demonstrate the power of EPG as a valuable tool in deciphering how viruses manipulate the behaviour of their insect vectors. These results are important in modelling acquisition and dispersal of viruses by aphid vectors. These findings will be valuable to the scholarship of molecular plant virology and plant-virus-insect interactions whose significance will increase as novel ways to control aphids and the viruses they spread are sought.

Reviewer #2: In regard of novelty, the study addresses a less well studied area where the authors investigated if a virus factor can modify vector fitness or behaviour. They chose CaMV P2 protein and Myzus persicae or the green peach aphid as CaMV vector. CaMV is a semi-persistently transmitted plant virus whereby it is not known if CaMV can have a direct involvement in modifying vector (aphid) behaviour, and its implication on onward transmission of the virus.

The strength of the study lies in the structured way they conducted their studies.

The group did prior work on CaMV , i.e.: Chesnais et al., 2020, 2021; Drucker et al., 2002 as referenced in the manuscripts. They discovered that P6-TAV, a multifunctional protein of CaMV, led to altered feeding behaviour of the aphids on Arabidopsis and this is likely to be mediated by indirect host-mediated effect as P6-TAV is involved mainly in modification of host plant physiology. CaMV P2 protein is a helper protein which has been shown to form a protein bridge between the aphid’s stylet and the virions. Hence the authors chose to investigate P2 protein in this study as a candidate for a virus-encoded factor which may modify vector behaviour.

Overall this paper is well laid out, where the authors investigated whether P2 protein of CaMV are involved in modifying aphid feeding behaviour as CaMV vector. The authors utilised wild-type CaMV virus (JI), mutant virus with abolished P2 (JIdP2), and a mutant virus whereby P2 has lost the ability to bind to the aphid mouthparts (JI-P2Rev5). The authors further tested whether the effect of P2 on aphid feeding behaviour is independent from other plant factors by using membrane feeding set up to allow aphids to feed directly on food solution with P2 virion, or P2:P3 virion complexes. The results did add new knowledge in the field.

The weakness of this study lies in the sentence structure and language, as well as there are additional experiments which can help clarify the results. There were several occasions where the choice of word and sentence structure need editing. The titles for the subsection in the results part of the manuscript also needs re-phrasing to increase its accuracy in reflecting the experimental result, and to avoid over assumption. Several statements are seemingly contradictory to one another, which can be due to poor sentence structuring and will need to be re-written to improve its clarity. The labelling on Figure 1 needs to be corrected (further details below). The suggested experiments are described in the following section.

Reviewer #3: This manuscript presents convincing evidence for a role of CaMV P2 in influencing aphid vector behaviour directly. The implication that a protein of a non-circulative virus manipulates the vector is highly significant and will be of broad interest to researchers in plant-pathogen-vector interactions. The experimental approaches are rigorous and nicely combine EPG recording with use of mutant viruses and an artificial membrane feeding system to provide consistent and significant results. The findings are novel and the work is suitable for publication following some minor changes.

**Part II – Major Issues: Key Experiments Required for Acceptance**

Reviewer #1: n/a

Reviewer #2: Fig 1b : can the authors check if the labelling is correct on the westen blot? If anti P2 is an antibody specific for P2 protein, then why does bands appear on JIΔP2 samples whereby P2 were not supposed to be present? Please verify.

Have the authors tested whether the aphids were able to acquire the mutant CaMV JI-P2Rev5 virus?

The authors did demonstrate that turnip plants were infected with CaMV JI-P2Rev5 virus, but with P2 being unable to bind to the aphid stylet – have the authors checked whether aphids can acquire and subsequently transmit CaMV JI-P2Rev5 mutant virus? Figure 4 did show that aphids fed on JI-infected plants performed significantly more intracellular punctures than aphids fed on CaMV JI-P2Rev5 infected plants. Whilst Figure S4 showed no significant difference between aphid feeding behaviour when different parameters were measured than Fig 4 (sap ingestion and time to first ingestion) using EPG on JI-infected and JI-P2Rev5 plants.

For L. 230 (Figure 3) : Did the authors detect P2 and/or P3:virions titer or verify its presence in the individual aphids after they were fed on the artificial membrane containing P2 and P3:virions? this is to verify that the effects observed

on the aphids were genuinely due to the P2 and/or P3 being taken up by the aphids and not just due to aphids experiencing feeding difficulties on the membrane for example? . Figure S2 showed viral components used for aphid feeding assay on the artificial medium, but is this is measured directly from the artificial medium itself and not from the individual aphids feeding on it? Figure S5(c) showed SDS PAGE gel of Sf9 crude extracts from the artificial medium, has any been done from the aphids itself?

L. 391 and L. 403: This paragraph gives out two seemingly contradictory statements which reduces clarity. In L. 391, the authors states “Our membrane feeding assay support the direct effect of P2”, whilst L. 403 reads as follows “Therefore, although the membrane feeding experiments do show a direct effect of P2 on aphids, interpretation of the results from the membrane feeding assays are difficult”. I prefer the more cautious way the authors framed their results here in L. 403, therefore can L. 391 please be re-phrased or rewritten?

Reviewer #3: The correlation analysis (presented in Figure 2c) provides a tantalising hint that P2 not only modifies aphid behaviour on virus-infected plants, but may also be involved in manipulating behaviour on healthy plants that are subsequently encountered and probed. Have the authors also considered investigating a correlation between number of punctures on healthy plant (the same data plotted here) against number of intracellular punctures on the virus-infected acquisition plant? Since P2 is acquired during intracellular punctures (epidermal / mesophyll cells), this approach may reveal a more biologically meaningful effect of P2 through the comparison between the three treatment groups. I therefore recommend that this additional analysis is carried out and included in the manuscript.

**Part III – Minor Issues: Editorial and Data Presentation Modifications**

Reviewer #1: There is clarification required in the statistical analyses. For example, the authors used Tukey for pairwise comparisons. However, I did not find any p values for this in the supplementary files, especially where differences were significant.

Figure 2 Panel ‘b’. It seems implausible that the pairwise comparisons for ‘Intracellular punctures during first penetration’ are not significant for Mock vs the mutant JIDP2. What was the p-value?

Figure 3 ‘penetration before first ingestion’ and ‘intracellular punctures’ What are the p values for the pairwise comparisons between the 3 and 5 treatments, respectively?

Figure 4: Panel ‘c’ ‘intracellular punctures’ What are the p values for the pairwise comparisons between the five treatments?

Line 328 – do you mean ‘less’ in place of ‘smaller’?

Reviewer #2: Please see more detailed comments below.

The title can be edited and made more concise? Current title is “The transmission helper protein of a plant virus….”. I would suggest to mention the name of the helper protein and the virus on the title directly.

L. 35: ‘first plant contact’ ? First contact with the plant

L. 38: ‘virus’ would be a more fitting choice of word here rather than ‘parasite’

L. 48: ‘instantly’ might be going too far here, as it was not shown here whether it is ‘instantenous’ or not. Please re-phrase

L.49: ‘prepares’ here personified CaMV, can this sentence be re-phrased? It is not accurate to assume that ‘CaMV prepares for..’ as what this sentence seems to be implying.

L. 50-51: This sentence is not clear. What does the authors mean by ‘minimizes vector implication’? Were the authors trying to say that it minimises the risk of non-transmission by the vector? Please re-phrase.

Introduction

The introduction started with introduction to plant viruses and their mode of transmission. Followed by examples of research on virus factors which mediates changes in vector fitness. Some relevant references in this subject area are missing and should be included. Certain references displayed here are not the most accurate references available to support the statement made and ideally should be edited. Please see comments below.

L. 58: ‘their’ morphology should be ‘the stylet’s morphology’. Not the morphology of the entire aphid as an insect as what the sentence reads as at the moment.

L. 70-74: Ray and Casteel 2022 is a review paper which focused more in depth about the role of effectors to mediate plant-virus-vector interaction. The statement on L. 70 whereby “the viral factors responsible for changes in vector fitness are still largely unknown” is not entirely accurate as more has been known about viral factors affecting vector fitness. Other references are more fitting to support the statement whereby particular viral factors have been identified which are responsible to change sin vector fitness , for example in Ziebell et al., 2011; Westwood et al., 2013; Shi et al., 2016; Wu et al., 2017; Tungadi et al., 2020; Arinaitwe et al., 2022.

L. 76: Donelly et al. 2019 is a paper on epidemiological modelling. Their work is still very much relevant to the theme of this manuscript. However in this manuscript, it was cited as though as it was a lab experiment paper. Donelly et al 2019 generated a modelling framework on non-persistent virus transmission where it includes aphid feeding behaviour, wingless or winged aphids, and inhibition of aphid settling on plants as parameters.

L. 87: ‘or not a’ ? unfinished sentence?

L. 102: will explore

L. 104: ‘ingest actively some cytosol’ does not sound right, please re-phrase this sentence.

L. 104: ‘if the plant is accepted’ , please re-phrase this sentence as well.

L.106-107: ingest phloem sap passively and continuously.

L. 102-111 : This paragraph can be shortened. I understand that the authors are trying to describe the sequence of events that occurred after the aphid lands on the plant which is relevant to the paper.

The authors then settled to choose P2 to test their hypothesis whether a non-circulative virus like CaMV can encode factors having a direct effect on the vector.

L. 121-122 : Why only P2 and not P3 as well?

Results

L.124: The title of this subsection does not really make sense, particularly in the part of “P2 probing”. Please re-phrase.

Fig 1a : can the authors add labelling on the figure itself please?

Fig 1b : can the authors check if the labelling is correct on the westen blot? If anti P2 is an antibody specific for P2 protein, then why does bands appear on JIΔP2 samples?

The main take home message is number of intercellular punctures made by the aphids increased significantly during JI (wild-type) virus infection but not on plants infected by JIΔP2 virus.

L. 148 : The authors then went on to test the effect of P2 on aphids using wild-type CaMV and P2 deletion mutant, JIΔP2. Deletion of P2 does not affect infectivity of the virus.

They did EPG and found that deletion of P2 also does not affect on phloem sap ingestion by the aphids.

However the number of intracellular punctures were a lot higher on JI-infected plants than JIΔP2 or mock inoculated ones. This section was summarised nicely on L.179-182 where they stated that CaMV infection (with and without P2) did increase phloem feeding and reduce the time taken by the aphids to reach the phloem.

L. 178-182: They also stated that the number of intercellular punctures were higher on aphids fed on wild-type CaMV Ji virus but not on the mutant virus or mock inoculated plants. The closing statement from this section where the authors indicated that ‘P2 was responsible for this’ sounds premature from the conclusion of this work to date. This work here does not exclude the possibility that the observed effects can be the due to alteration in the host plant, thus an indirect effect of CaMV.

Have the authors tested whether aphids were able to acquire the mutant CaMV JI-P2Rev5 virus? The authors did demonstrate that turnip plants were infected with CaMV JI-P2Rev5 virus, but with P2 being unable to bind to the aphid stylet – have the authors confirmed that aphids can acquire CaMV JI-P2Rev5 ? i

L. 183: The title for this subsection is not entirely accurate. Please re-phrase and be more specific, i.e: mention P2 protein on the title?

L. 227 – 228: The title for this subsection on “Direct interaction between P2 and Myzus persicae modifies the probing behaviour of the aphid vector” needs to be re-phrased as the authors have not shown protein-protein interaction between P2 and aphid stylet here in their results. It would be more accurate to re-name the subsection as “Acquisition of P2 virions (and/or P3 virions) xxx by M. persicae modifies xxx ”.

L. 265-266: Similar to my previous comment. The title for this subsection on ‘The interaction between P2 and the aphid stylets is required for changing probing behaviour’ needs to be re-phrased as the authors have not shown a direct protein-protein interaction here between P2 and the aphid stylets. It would be more accurate to re-name the subsection as “Acquisition or feeding from xxx “.. or “Aphids feeding on mutant virus with abolished function of P2 interaction to xxxx “.

L. 287: ‘interacting with intestine’? please re-phrase. Did the authors meant to say interacting with host factors in the aphid gut/intestine?

Figure 2 : has the authors showed here that aphids can acquire JIΔ2P?

L. 301: performed more than twice

L. 315: the authors did acknowledge that ‘the acrostyle-binding capacity for P2 is important for this’. Hence I think it is important that the authors demonstrate whether aphids can acquire and subsequently able to transmit CaMV JI-P2Rev5 from a CaMV JI-P2Rev5 infected plants or not. This is important as otherwise, the observed aphid behaviour when feeding on CaMV JI-P2Rev5 infected plants will be due to an indirect host-mediated effect whether the aphids were responding to virus-mediated effect to the plant hosts.

L. 340-341: Is there any reference that can be cited which shows that phloem contained higher virus titer than the surrounding single cells?

L. 338-345: This paragraph is confusing, does the authors meant to say that P2 is not involved in the alterations of aphid behaviour (i.e: lower number of cell punctures) as they observed a similar effect on aphids fed on both wild-type CaMV inoculated plants and CaMV P2 deletion mutant inoculates plants? This somewhat contradicts the results from Fig.1 where on L. 172-175, it states that ‘we observed P2-speficic alterations for the number of intracellular punctures during the first probe which was twice as high on JI-infected plants than on JI P2-deletion mutant infected plants’. This is also re-iterated again on L. 347-348. Please edit this paragraph.

L. 347: Is there more than one CaMV isolate used in this study?

L 357 : Has there been studies to show whether the level or titer of P2 protein on the stylet remained the same throughout ? whether aphids for example had higher titer of P2 protein on their stylet tips during the initial acquisition, which then increased and tapered off?

L.364-367 contradicts L. 338-339, please check and edit.

L. 389: The opening sentence where “We show here that P2 changes aphid probing behaviour..” again does not sit comfortably with me as the authors have not shown yet whether the effect was direct or indirect. Rather than addressing the ‘direct or indirect’ effect in the second sentence, it would be better if the authors opens this paragraph by stating the fact directly “We show here that P2 may directly or indirectly changes…….”

L. 391 and L. 403: This paragraph gives out two seemingly contradictory statement which reduces clarity. In L. 391, the authors states “Our membrane feeding assay support the direct effect of P2”, whilst L. 403 reads as follows “Therefore, although the membrane feeding experiments do show a direct effect of P2 on aphids, interpretation of the results from the membrane feeding assays are difficult”. I prefer the more cautious way the authors framed their results here in L. 403, therefore can L. 391 please be re-phrased. Did the authors measure the success rate of subsequent virus inoculation onto healthy plants by the aphids after being fed on the membrane? Did the authors also detect and measure P2 and/or P3 protein presence on the aphids after they were fed on the membrane to verify that the effects observed were genuinely due to the P2 and/or P3 proteins being taken up by the aphids and not just due to aphids experiencing feeding difficulties on the membrane.

L. 175 – 176 and L. 339-342 : Both set of sentences also seems to convey contradictory message whereby on L. 175-176, the authors stated that the number of intracellular punctures during the first probe was twice as high on JI-infected plants than on JIdP2 infected ones or mock-inoculated ones. This seemingly contradict the statement on L. 339-344 where the authors stated that P2 plays no role in this as the observed the same behaviour on plants infected with wild-type virus and on plants infected with JIdP2.. whereby “this” was mentioned in L. 339 as “lower number of test punctures combined with the longer phloem ingestion on infected plants”. Implying that on L. 175 : aphids made more punctures on JI-infected than on JIdP2 infected plants, however on L. 339-342 : aphids showed similar number of test punctures on both JI- and JIdP2 infected plants.

This is either a grammatical error or the sentences need to be restructured as to avoid confusion to readers.

Reviewer #3: I have a general issue with the term ‘feeding behavior’ (e.g. in the heading line 124 and numerous other places in the text) to indicate all behaviours displayed by aphids during stylet penetration. On plants, only E2 (sustained phloem sap ingestion) is really “feeding”. Better to use more generic (‘stylet penetration’ / ‘probing’) alternatives throughout the manuscript.

The authors seem to over-state the evidence for causative links in some parts of the text. I have suggested more cautious language in parts, and made some other specific comments with line numbers indicated as follows.

Line Comment

50 The meaning of ‘minimises vector implication’ is not clear, what is the significance?

56 ‘Hemiptera’ is an order (so not italics)

94 ‘fixation’ implies a non-reversible link - change to ‘adherence’

108, 110 ‘tissue cells’ – can a more useful and specific term be used? Peripheral? Phloem sieve elements are part of a (phloem) tissue, so ‘tissue and from phloem sap’ does not make sense

125, 158 The statement does not agree with Figure 1 b), which indicates that P2 is not detected in WT plants – has the figure been mislabelled?

125 Figure 1 c) & d) and subsequent figures need full axis labels – presumably duration (hours) and occurrence (number of events per recording)? Note spelling of ‘occurrence’ (two r’s). What is being plotted – means and standard errors? This also needs clarifying in subsequent figures.

160 Change ‘taking’ to ‘taken’

167 Change ‘1a’ to ‘1c’

178 Change to ‘….without P2) was associated with an increase in the duration….’ (there’s no basis for a causal link at this point).

179 ‘reduce the time required to reach it’ is not supported by the data and should be deleted, unless it can be evidenced. Figure 1c) shows no significant difference (time to first sap ingestion).

182 Change to ‘… was associated with this.’ (again, no basis for a causal link at this stage).

187 Change ‘lists’ to ‘presents’.

215 Change ‘was’ to ‘may be’.

287 I suggest changing ‘intestine’ to ‘more posterior parts of the alimentary canal’ (or something similar). There seem to be lots of other possibilities for potential interaction sites (other than the intestine).

306 Change ‘only on the rim of significance’ to ‘marginally not significant’.

315 Change ‘this’ to ‘the observed changes in vector behavior’ (or similarly expand this rather brief last sentence).

316 Delete ‘a’

326 Change ‘behavior’ to ‘behavioral’.

328-330 Can these statements (‘aphids localize the phloem faster’/’faster access to the phloem’) be supported by reference to specific parameters and statistical outcomes? Time to first E1 / E2?

352 Change ‘is’ to ‘may be’.

362 Change ‘did’ to ‘exhibited’.

393 Change to ‘altered subsequent probing…’

433 Suggest changing to ‘is not definitive evidence’. Otherwise this sentence does not make sense, particularly when the next sentence allows for other hypotheses.

434 Again, it seems strange to highlight particular regions of the alimentary canal (cibarium, intestine) when many other potential binding sites are possible.

454 Change ‘inversed’ to ‘distinctly different’.

502 Change ‘depression’ to ‘negative air pressure’.

509 Change aphids to aphids’.

556, 566, 586 Change ‘in’ or ‘by’ to ‘previously’.

PLOS authors have the option to publish the peer review history of their article (what does this mean?). If published, this will include your full peer review and any attached files.

Reviewer #1: **Yes: **Francis Wamonje

Reviewer #2: No

Reviewer #3: No

Figure Files:

Data Requirements:

Reproducibility:

References:

---

## [Editor Report · Decision Letter 1]

27 Jan 2023

Dear Dr Drucker,

We are pleased to inform you that your manuscript 'The cauliflower mosaic virus transmission helper protein P2 modifies directly the probing behavior of the aphid vector Myzus persicae to facilitate transmission' has been provisionally accepted for publication in PLOS Pathogens.

Best regards,

John P Carr

Guest Editor

PLOS Pathogens

Shou-Wei Ding

Section Editor

PLOS Pathogens

Kasturi Haldar

Editor-in-Chief

PLOS Pathogens

orcid.org/0000-0001-5065-158X

Michael Malim

Editor-in-Chief

PLOS Pathogens

orcid.org/0000-0002-7699-2064
---

## [Editor Report · Acceptance letter]

2 Feb 2023

Dear Dr Drucker,

We are delighted to inform you that your manuscript, "The cauliflower mosaic virus transmission helper protein P2 modifies directly the probing behavior of the aphid vector *Myzus persicae* to facilitate transmission," has been formally accepted for publication in PLOS Pathogens.

Best regards,

Kasturi Haldar

Editor-in-Chief

PLOS Pathogens

orcid.org/0000-0001-5065-158X

Michael Malim

Editor-in-Chief

PLOS Pathogens

orcid.org/0000-0002-7699-2064